# OFFLINE COMMUNICATION LEARNING WITH MULTI-SOURCE DATASETS

## ABSTRACT

Scalability and partial observability are two major challenges faced by multi-agent reinforcement learning. Recently researchers propose offline MARL algorithms to improve scalability by reducing online exploration cost, while the problem of partial observability is often ignored in the offline MARL setting. Communication is a promising approach to alleviate the miscoordination caused by partially observability, thus in this paper we focus on offline communication learning where agents learn from an fixed dataset. We find out that learning communications in an end-to-end manner from a given offline dateset without communication information is intractable, since the correct communication protocol space is too sparse compared with the exponentially growing joint state-action space when the number of agents increases. Besides, unlike offline policy learning which can be guided by reward signals, offline communication learning is struggling since communication messages implicitly impact the reward. Moreover, in real-world applications, offline MARL datasets are often collected from multi-source, leaving offline MARL communication learning more challenging. Therefore, we present a new benchmark which contains a diverse set of challenging offline MARL communication tasks with single/multi-source datasets, and propose a novel Multi-Head structure for Communication Imitation learning (MHCI) algorithm that automatically adapts to the distribution of the dataset. Empirical result shows the effectiveness of our method on various tasks of the new offline communication learning benchmark.

## 1 INTRODUCTION

Cooperative multi-agent reinforcement learning is essential for many real-world tasks where multiple agents must coordinate to achieve a joint goal. However, the problems of scalability and partial observability limit the effectiveness of online MARL algorithms. The large joint state-action space makes exploration costly especially when the number of agents increases. On the other hand, the partial observability requires communication among agents to make better decisions. Plenty of previous works in MARL try to find solutions for these two challenges, with the hope to make cooperative MARL applicable to more complicated real-world tasks (Sukhbaatar et al. (2016); Singh et al. (2019); Yang et al. (2021)).

Recently, emerging researches apply offline RL to cooperative multi-agent RL in order to avoid costly exploration across the joint state-action space, thus the scalability is improved. Offline RL is defined as learning from a fixed dataset instead of online interactions. In the context of single-agent offline RL, the main challenge is the distributional shift issue, which means that the learned policy reaches unseen state-action pairs which are not correctly estimated. By constraining the policy on the behavioral policy, offline RL has gained success on diverse single-agent offline tasks like locomotion (Fu et al. (2020)) and planning without expensive online exploration(Finn & Levine (2017)).

As the problem of scalability can be promisingly alleviated by utilizing offline datasets, another challenge of MARL, i.e., the partial observability, can be addressed by introducing communications during coordination. Communication allows agents to share information and work as a team. A wide range of multi-agent systems benefit from effective communication, including electronic games like

StarCraft II (Rashid et al. (2018) and soccer (Huang et al. (2021)), as well as real-world applications, e.g., autonomous driving (Shalev-Shwartz et al. (2016)) and traffic control (Das et al. (2019)).

Although several offline MARL algorithms (Yang et al. (2021); Pan et al. (2022)) have been proposed to tackle with the scalability challenge recently, how to deal with the partial observability in the offline MARL setting, has not received much attention. Unluckily, simply adopting communication mechanism in offline MARL, i.e., learning communication in an end-to-end manner by offline datasets is still problematic. Finding effective communication protocols without any guidance can be the bottleneck especially when the task scale increases. It may converge to sub-optimal communication protocols that influence the downstream policy learning. To handle this problem, in this paper, we investigate the new area of offline communication learning, where the multiple agents learn communication protocols from a static offline dataset containing extra communication information. We denote this kind of dataset as "communication-based dataset" to distinguish it from single-agent offline dataset. In real-world applications, communication-based dataset may be collected by a variety of existing communication protocols, like handcraft protocols designed by experts, or hidden protocols leaned by other agents. Therefore, communication-based dataset can be established in offline MARL learning to boost the performance of the downstream tasks.

Previous offline RL works focus on eliminating the problem of distributional shift, while offline MARL communication learning faces different challenges. Unlike policy learning which is directly guided by reward signals since actions influence the expected return, it is hard to evaluate communication learning since communication serves as an implicit impact between agents . What's worse, it is likely that the offline dataset is multi-source in real world, thus trajectories may be sampled by different communication protocols as well as policies. The multi-source property introduces extra challenges as we cannot simply imitate the dataset communication. Offline communication learning algorithms need to distinguish the source of each trajectories before learning from them. In this paper, We propose Multi-head Communication Imitation (MHCI) that accomplishes multi-source classification and message imitation at the same time. To our best knowledge, MHCI is the first to learn a composite communication from a multi-source communication-based dataset. We also provide theoretical explanation on its optimality under the dataset.

To better evaluate the effectiveness of our algorithm as well as for further study, we propose an offline communication learning benchmark, including environments from previous works and additional environments that require sophisticated communication. The empirical results show that Multi-head Communication Imitation (MHCI) successfully combines and refines information in the communication-based dataset, thus obtains outperformance in diverse challenging tasks of the offline communication learning benchmark.

Our main contributions are two-folds: 1) we analyze the new challenges in offline communication learning, and introduce a benchmark of offline communication learning which contains diverse tasks.; 2) we propose an effective algorithm, Multi-head Communication Imitation (MHCI), which aims to address the problem of learning from single-source or multi-source datasets, and our method shows superior outperformance in various environments of our benchmark.

## 2 RELATED WORKS

**MARL with communication**   Multi-agent reinforcement learning has attracted great attention in recent years. (Tampuu et al. (2017); Matignon et al. (2012); Mordatch & Abbeel (2017); Wen et al. (2019)) In MARL, the framework of centralized training and decentralized execution has been widely adopted (Kraemer & Banerjee (2016); Lowe et al. (2017)). For cooperative scenarios under this framework, COMA (Foerster et al. (2018)) assigns credits to different agents based on a centralized critic and counter-factual advantage functions, while another series of works, including VDN (Sunehag et al. (2018)), QMIX (Rashid et al. (2018)) and QTRAN (Son et al. (2019)), achieve this by applying value-function factorization. These MARL algorithms show remarkable empirical results when tested on the popular StarCraft unit micromanagement benchmark (SMAC) (Samvelyan et al. (2019)).

CommNet (Sukhbaatar et al. (2016)), RIAL and DIAL (Foerster et al. (2016)) are seminal works that allow agents to learn how to communicate with each other in MARL. CommNet and DIAL design the communication structure in a differentiable way to enable end-to-end training, and RIAL trains

communication with RL algorithms by constraining communication to be discrete and regarding it as another kind of actions. To make communication more effective and efficient, IC3Net (Singh et al. (2019)), Gated-ACML (Mao et al. (2020b)) , and I2C (Ding et al. (2020)) utilize a gate mechanism to decide when and who to communicate with. TarMAC (Das et al. (2019)) and DAACMP (Mao et al. (2020a)) achieve targeted communication by introducing attention mechanism. NDQ (Wang* et al. (2020)) uses information-theory-based regularizers to minimize communication. MAIC (Yuan et al. (2022)) realizes sparse and effective communication by modeling teammates.

**Offline MARL**   Offline RL is a hot topic in the last few years. It focuses on RL given a static dataset. The problem of distributional shift is critical for offline RL (Fujimoto et al. (2019)), and typical solutions include constraining the policy (Kumar et al. (2019); Wu et al. (2020)) or the Q value (Kumar et al. (2020)). Recently, offline MARL is also attracting interests. As a combination of two areas, offline RL and MARL, it faces new challenges of larger state-action space. Yang et al. (2021) proposes a even more conservative method compared to single-agent offline RL. Pan et al. (2022) finds that conservative offline RL algorithms converge to local optima in the multi-agent setting, thus proposes better optimization methods.

**Multi-source fusion**   Works in other fields also concern about multi-source fusion. For example, Sun et al. (2021) use an attention mechanism to fuse multiple word features for downstream tasks.

## 3   BACKGROUND

In a fully cooperative multi-agent RL, the environment is generally modelled as a Dec-POMDP Oliehoek & Amato (2016). Adding communication, Dec-POMDP with communication is defined as $G =< I, S, A, P, R, \Omega, O, n, \gamma, M >$. $I = \{1, 2, \cdots, n\}$ stands for the set of the agents. $s \in S, a_i \in A, o_i \in \Omega$ are the state, action and observation of a certain agent, and the observation of agent $i$ $o_i$ is calculated by the observation function $o_i = O(s, i)$. $P(s'|s, a), R(s, \mathbf{a})$ are the transition function and the reward function, and $\gamma \in [0, 1)$ is the discount factor. When communication is added to this model, $m_{ij}$ is denoted as message conveyed from agent $i$ to agent $j$. In practice, we take $s = o_:$ by default. The offline dataset is denoted as $\mathcal{D}$.

Based on the basic definitions of Dec-POMDP with communication, we can further denote the action-observation history $\tau_i \in T \equiv (\Omega \times A)^*$, and a policy $\pi_i(m_{:i}, \tau_i)$ is defined on the history. In order to unify the notation of communication protocols proposed by previous works, we first give a general form of communication

$$m_{ij} = C_{ij}(o_i), a_j = \pi_j(m_{:j}, o_j, \tau_j) \, (\forall i, j \in I), \tag{1}$$

where $C_{ij}$ is an arbitrary communication function, and : is the abbreviation of all the agents. Although $o_j$ is included in $\tau_j$, here we abuse the notation for convenience. In Equation 1, each message from $i$ to $j$ is generated from the communication function, and an action is determined through the policy function $\pi$, by taking as input all the received messages, individual observation, and individual history if needed. In Appendix A, we list the expression of the communication protocols in previous works. Generally speaking, they take different inductive bias into account and can all be viewed as modified protocols from Equation 1.

## 4   A MOTIVATIVE EXAMPLE

Before digging into the specific method of offline communication learning, we first illustrate why learning communication directly from communication-based dataset is important, rather than learning in an end-to-end manner, i.e., learning communication from scratch. In this section, we analyze the influence of the number of agents $n$ and the dimension of states $p$, by comparing the performance in an imaginary communication game with tunable $n, p$. The number of agents corresponds to the scalability of a task, and the dimension of states reflects the complexity of information sources (e.g. images are complex, while velocity and acceleration information isn't).

The imaginary communication game consists of $n$ agents, with state dimension $p$ and observation dimension $q$. And **$2q$** (not $q$) dimensions of the state space are related to the policy of each agent in (4), therefore besides the $q$ dimensions that can be directly observed as defined in (3), the other $q$

dimensions are not observed thus come from communication. The max horizon $T = 1$, and each of the $p$-dimension states is randomly initialized from $-1, 1$. Details are in Appendix C.

$$T = 1, \ s \in \{-1, 1\}^p \tag{2}$$

$$o_i = (s_{index_1^{(i)}}, s_{index_2^{(i)}}, \cdots, s_{index_q^{(i)}}) \tag{3}$$

$$a_i^* = \begin{cases} 1 & \prod_{j=1\cdots 2q} s_{index_j^{(i)}} > 0 \\ 0 & otherwise \end{cases} \quad (a_i^* \text{ is the optimal action of agent } i) \tag{4}$$

$$r = \sum_{i=1\cdots n} \mathbb{I}(a_i = a_i^*). \tag{5}$$

Figure 1 shows how the performance decays with the increasing $n$ and $p$. In general, learning communication from dataset performs better than learning communication from scratch, especially when the task becomes harder with a larger $n$ or $p$. The basic reason is that, in challenging tasks where the whole state space is enourmous, pure offline MARL methods still have difficulty in finding unobserved information related to an optimal policy, even if hard exploration in the state space is no longer a problem. A randomly initialized communication function may converge to sub-optimal, that confuses the downstream policy to some extent. For example, in a multi-agent navigation task with image inputs, the agent needs to communicate whether the goal is in sight. However, there's too much redundant information in the image, so the communication tends to converge to easier patterns like the color or background of the goal, which are misleading in downstream policy learning.

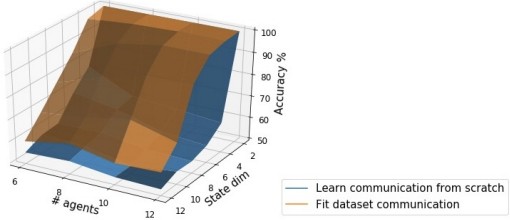

Figure 1: Evaluation accuracy with different number of agents $n$, and different dimension of the state space $p$, comparing fitting the dataset communication and learning communication from scratch.

## 5 METHOD

In this work, we propose Multi-head Communication Imitation (MHCI) for offline communication learning as shown in Figure 2. In Section 5.1, we provide the complete definitions of the multi-source communication in the offline setting, and derive the offline optimality according to the communication in the dataset. Finally in Section 5.2, the full structure of our Multi-head Communication Imitation (MHCI) is introduced, which learns a composite communication protocol that supports the optimal policy according to the offline dataset.

### 5.1 UNIVERSAL AND LOCAL COMMUNICATION IN THE OFFLINE SETTING

In order to introduce the concept of Universal and Local Communication, we first define how the multi-source dataset is split. Denote a communication function as $C(o_{\cdot}) : \Omega^n \to \mathbb{R}^{p \cdot n^2}$ ($p$ is the dimension of each message $m_{ij}$), which specifically is

$$C((o_1, \cdots, o_n)) = \\ (C_{11}(o_1), C_{12}(o_1), \cdots, C_{1n}(o_1), C_{21}(o_2), \cdots, C_{2n}(o_2), \cdots, C_{n1}(o_n), \cdots, C_{nn}(o_n)). \tag{6}$$

Assume that the communication in dataset $\mathcal{D}$ can be discriminated by source into $|G|$ groups, with state domain and communication function

$$G = \{(S^{(1)}, C^{(1)}(\cdot)), (S^{(2)}, C^{(2)}(\cdot)), \cdots, (S^{(|G|)}, C^{(|G|)}(\cdot))\}. \tag{7}$$

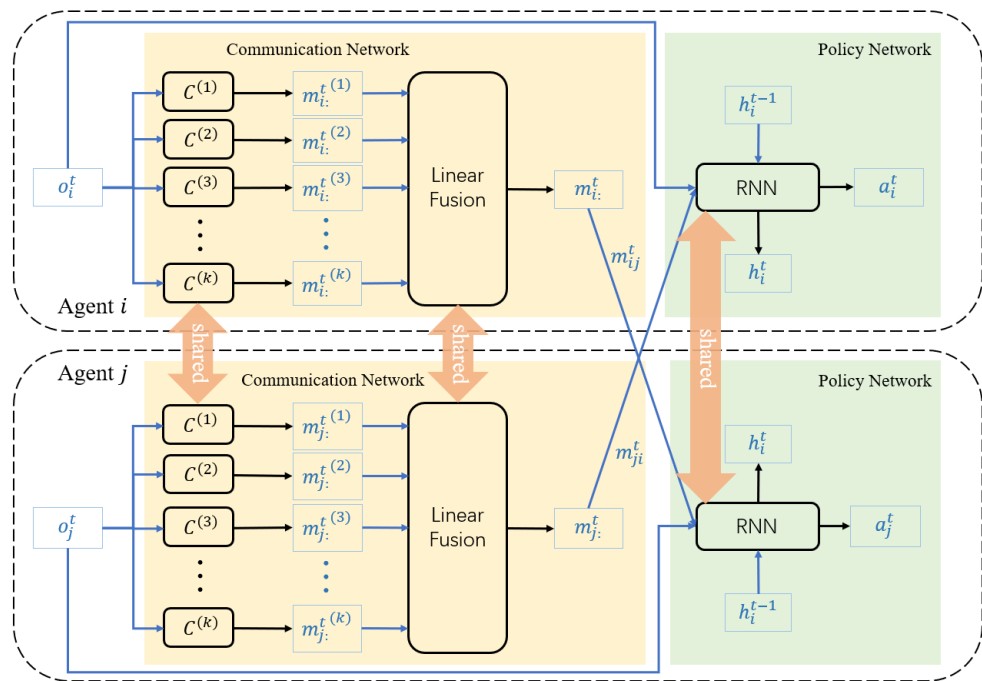

Figure 2: The pipeline of offline communication learning, breaking into two parts, the communication network and the policy network. Our work focuses on offline communication learning using a multi-head structure shown in the communication network part. The messages are generated by first computing the messages in all the heads, and then a linear fusion is applied on them.

We name each communication function $C^{(i)}, i \in [|G|]$ as one **Local Communication (LC)**. There are $|G|$ Local Communications in the dataset in total.

**Definition 1.** *The relation that $C^{(1)}$ **dominates** $C^{(2)}$ is defined as*

$$\exists f, s.t. \ \forall j, k \in [n], f_{jk}(C_{jk}^{(1)}(o_j)) = C_{jk}^{(2)}(o_j). \tag{8}$$

**Definition 2.** *Define the **Universal Communication (UC)** as the communication function that contains all of dataset communication function.*

$$UC : S \rightarrow \mathbb{R}^*,$$

$$\exists f_1 \cdots f_{|G|}, \forall i \in [|G|], j, k \in [n], f_{jk}^{(i)}(UC_{jk}(o_j)) = C_{jk}^{(i)}(o_j)$$

$$i.e.$$

$$\forall i \in [|G|], UC \text{ dominates } C^{(i)}.$$

$$\tag{9}$$

Compared to the Local Communication, the Universal communication includes the information of all the Local Communications $C^{(1)}, C^{(2)}, \cdots, C^{(|G|)}$.

After introducing the concept of Universal and Local Communication, we prove that the Universal Communication is sufficient for obtaining policies that match or outperform the dataset in Theorem 1. For simplicity, first denote the optimal policy and the optimal value function under a Dec-POMDP $T_C(M)$ (transformed from the real MDP $M$ by the communication function $C(\cdot)$) as $\pi_{T_C(M)}^*, V_{T_C(M)}^*$.

**Theorem 1.** *The optimal expected return based on the Universal Communication is greater than or equal to that of dataset communications and policies, i.e.,*

$$\forall (S, LC, \pi) \in G_\pi, V_{T_{UC}(M_{\mathcal{D}})}^*(s_{init}) \geq V_{T_{LC}(M_{\mathcal{D}}), \pi}(s_{init}). \tag{10}$$

Theorem 1 means that the optimal policy given $UC(\cdot)$ is always equal to or better than that of the dataset, under the dataset MDP $M_{\mathcal{D}}$ as defined in (Fujimoto et al. (2019)). The whole proof is in appendix B.

In fact, the provably true Universal Communication is at least the concatenation of all Local Communications, because without any additional assumptions on communication, the local communications may include distinct information. As a result, we are actually approximating the Universal Communication in the following algorithm design.

## 5.2 MULTI-HEAD COMMUNICATION IMITATION

In section 5.1, we have shown that in order to guarantee optimality compared with the performance of dataset policies, combining all the information contained by dataset communication is essential for the policy training afterward. To this end, we propose a Multi-head Communication Imitation (MHCI) pipeline that learns communication from multi-source datasets. It simultaneously optimizes the predicted category of each communication and imitates the dataset communication. The learned multi-head communication can be viewed as Local Communication mentioned above. Besides training, we adopt a Linear Fusion module to fuse the learned communications in different categories. The fusing phase can be understood as approximating the Universal Communication. With MHCI and a policy learner, we're able to master the communication and policy in challenging tasks with multi-source offline datasets.

**Multi-head structure**  We design a multi-head stucture to classify the true category. The structure is inspired by the widely used attention module, where query and key are first computed from the input, and the weight of each element is obtained from the inner-product of key and query. In our multi-head structure, we simply compute the key from dataset communication, and the query from the observation. After taking the inner-product and applying the softmax operator, we get a prediction probability of each category, which is used in the overall imitation loss in Equation 11. In practice, the communication heads $C^{(1)}, C^{(2)}, \cdots, C^{(k)}$ share most networks except the last two layers for efficiency.

$$Loss_{Imitation} = \sum_{i=1}^{k} prob_i \cdot MSE(C^{(i)}(o_{\mathcal{D}}), m_{\mathcal{D}}). \tag{11}$$

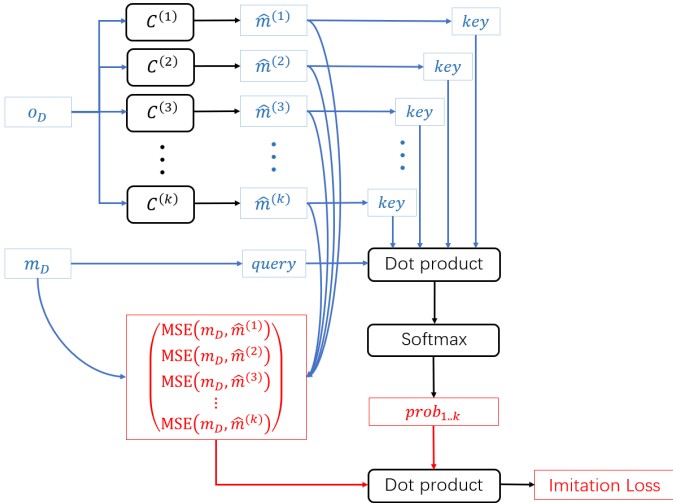

Figure 3: The schematics of the network structure and the loss computation in Equation 11.

**Linear fusion**  Although the Multi-head Communication Imitation successfully learns the communication function of each category, we still need to fuse all those categories of generated messages in the testing phase. A trivial fusion method is to concatenate them all, but it is bandwidth-consuming as the dimension of all the received messages $m_{\cdot i}$ is $k \cdot p$. To fuse different categories of message output, as long as not adding extra burden on computation, we use Linear fusion which is enough

in most cases empirically. It can be viewed as an approximation of the concatenated messages. We fuse the received messages in the following way.

$$m_{fusion} = \mathbf{A_{kp \times p}}(m^{(1)}, m^{(2)}, \cdots, m^{(k)})^T, \tag{12}$$

where $(m^{(1)}, m^{(2)}, \cdots, m^{(k)})$ are the learned messages with dimension $p$. $k$ is the total number of heads in MHCI. Further investigations on Linear Fusion are in the ablation study in Section 6.3.

In practice, the fusion matrix $A_{kp \times p}$ is learned by the downstream RL loss. Since it's probable that the dataset communication isn't optimal, we also assign an additional learned head of communication. So the optimization of the linear fusion is as follows

$$\mathbf{A}, \phi = \underset{A, \phi}{argmin}\, Loss_{RL}(o, a, \mathbf{A}(m^{(1)}, m^{(2)}, \cdots, m^{(k)}, C_\phi^{(k+1)}(o))^T). \tag{13}$$

## 6 EXPERIMENT

In order to evaluate offline communication learning algorithms, we introduce a benchmark that consists of many kinds of communication-intensive cooperative multi-agent tasks. Tasks introduced in previous researches are investigated in our work, and we also create new tasks that require sophisticated communication. The details of the benchmark are included in Section 6.1. All the environments except Room (an added environment introduced in Section 6.1) are built based on StarCraft II. It is a widely used platform in a variety of researches on MARL and communication (Samvelyan et al. (2019); Rashid et al. (2018); Wang* et al. (2020)). The implementation details including hyperparameters and dataset generating scheme are in Appendix D.1.

Besides the part of communication learning, we use Implicit Constraint Q-learning (ICQ, Yang et al. (2021)) as the downstream offline policy learning algorithm in our experiments. It is designed to trust only state-action pairs that appears in the dataset. Our experiment results in Section 6.2 are based on ICQ combined with relative offline communication learning methods. By keeping the downstream offline MARL algorithm all the same among comparison, we empirically show how communication learning influences the overall performance.

### 6.1 OFFLINE COMMUNICATION LEARNING BENCHMARK

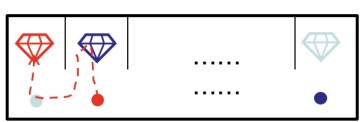

Figure 4: The environment *Room*

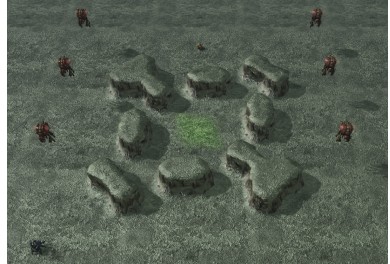

Figure 5: The environment *6o1b_vs_1r* in StarCraft II

Besides environments used in previous researches, the first added environment is *Room*. Each agent is corresponding to one of the goals (diamonds in Figure 4), and acquires reward only when it arrives at the proper goal of the same color. The goals are initialized randomly in the upper row, while the agents are initialized randomly in the lower row. The challenge comes from the fact that, each agent has no idea where the corresponding goal is. So the optimal joint policy is first going up to check the color of that diamond, and then sending the coordinate message to the agent with the same color. In this way, every agent is aware of its proper destination, and is able to head for the proper goal with reward.

We also come up with a coordination task in StarCraft II called 6o1b_vs_1r. In this environment, there are 6 observers who can't move, and 1 agent (between the two observers on the top) needs

to reach the enemy's position and kill it. The 6 observers and 1 agent are initiallized in the fixed position, while the enemy is randomly initialized in the lower-left or lower-right corner. However, each agent has a limited sight range. So the message is transferred from the lowest observer to upper observer, and finally transferred to the agent that can move to the enemy.

During designing new environments with sophisticated communication, we conclude the following patterns that make the task really challenging.

- **Inconsistent communication** It means that the communication is inconsistent among different receivers. Unlike broadcasting part of one's observation to everyone, sending the necessary message without redundant information is a more effective way.
- **Delayed-effect communication** Consider a special case that the optimal policy is based on the received message in the past. Optimizing in an environment with such communication pattern requires recursive backpropogating along the historic policy network (e.g. RNN), adding to additional difficulty in convergence.

In conclusion, there are different kinds of environments in the benchmark. Some come from previous online cooperative MARL researches, and others are new environments where a sophisticated communication is essential for solving these tasks. Specifically, we assign moderate-size datasets in order to investigate how offline communication learning helps downstream policy learning in challenging tasks. Although real-world tasks involve larger multi-agent systems and more complicated observations than experimental settings, we hope the offline communication learning benchmark can bridge the gap between experimental RL tasks and more challenging real-world applications.

## 6.2 MAIN RESULTS

|  | MHCI (Ours) | Learning communication from scratch | Pure imitation |
|---|---|---|---|
| 3b_vs_1h1m-medium-random | **38.1 ± 7.6** | 16.3 ± 4.8 | 25.7 ± 15.5 |
| 1o10b_vs_1r-medium-random | 52.2 ± 13.4 | 35.3 ± 1.9 | **60.4 ± 6.3** |
| 5z_vs_1ul-medium-random | 47.9 ± 23.0 | 25.0 ± 10.2 | 50.0 ± 13.5 |
| MMM-medium-random | **17.6 ± 2.9** | 5.7 ± 3.1 | 10.0 ± 1.2 |
| 1o2r_vs_4r-medium-random | 59.7 ± 29.5 | 65.3 ± 8.4 | 63.1 ± 15.2 |
| Room-medium | 82.5 ± 6.3 | 74.2 ± 3.5 | **95.4 ± 3.6** |
| Room-expert-expert | **86.3 ± 5.8** | 58.2 ± 0.4 | 58.7 ± 0.5 |
| Room-expert-expert-medium | **69.4 ± 6.7** | 59.7 ± 1.5 | 55.7 ± 2.5 |
| Room-medium-random | **67.2 ± 6.4** | 49.9 ± 1.7 | 61.0 ± 1.2 |
| 6o1b_vs_1r-medium-random | **45.0 ± 5.7** | 44.0 ± 6.5 | 44.0 ± 1.4 |

Table 1: The performance comparison of our method MHCI, learning communication from scratch and pure imitation, normalized by the score of the expert (from 0 to 100). The results are averaged over three random seeds with standard deviation.

As shown in Table 1, we compare our algorithm with two other communication learning strategies. Learning communication from scratch in the second column means that instead of using the dataset communication, the communication function is trained together with the policy in an end-to-end manner. Pure imitation means that without discriminating different sources, the communication is learned by minimizing the MSE loss between dataset and learned communication. The table is split into two parts, including environments introduced in previous works and added environments respectively.

In the first part (the first 5 rows of results), we cope with StarCraft II tasks introduced in Wang* et al. (2020). From the results, we can conclude that by learning from communication-based datasets, both MHCI and pure imitation perform better in 3b_vs_1h1m-mdeium-random, 5z_vs_1ul-medium-random and MMM-medium-random. Among the 3 methods, MHCI takes the property of multi-source into account and achieves higher scores than Pure imitation. In 1o10b_vs_1r-medium-random, pure imitation performs better than MHCI and learning from scratch, which is probably because the large number of agents (11 agents in total) makes multi-head learning difficult. In 1o2r_vs_4r-medium-random, all the three strategies actually perform similarly.

The comparison in the added environments described in Section 6.1 is included in the second part (the last 5 rows of results). In the single-source dataset Room-medium, pure imitation obtains the highest score, and MHCI is the second. We also compare under three mixing schemes in *Room*. The results show that our algorithm outperforms learning communication from scratch by using dataset communication, as well as pure imitation, which neglects the multi-source property. In 6o1b_vs_1r where a cascading communication is required, all the three methods fail to have good scores. We look forward to future algorithms that can handle 6o1b_vs_1r-medium-random.

In conclusion, we conduct comparison among the three methods, MHCI, learning communication from scratch and pure imitation, on the created benchmark of offline communication learning. In this way, we show that by utilizing dataset communication, both MHCI and pure imitation work better than learning communication from scratch. And in many datasets, dealing with the multi-source property gives MHCI an additional advantage over pure imitation. Therefore, we can draw the conclusion that MHCI boosts downstream policy learning in offline communication learning with multi-source datasets.

## 6.3 ABLATION STUDY

In this section, we look into the effectness of each module in MHCI. As introduced in Section 5, our method includes two main parts, the multi-head structure for communication imitation learning, and the way of fusing the messages from different sources. Therefore, we respectively compare the modules with other alternatives.

| | MHCI (ours) | closest | |
|---|---|---|---|
| Room-expert-expert | $\mathbf{86.3 \pm 5.8}$ | $79.2 \pm 7.3$ | |
| 3b_vs_1hm-medium-random | $\mathbf{38.1 \pm 7.6}$ | $1.7 \pm 1.6$ | |
| | Learned linear fusion (Ours) | PCA-based fusion | Hidden |
| Room-expert-expert | $\mathbf{86.3 \pm 5.8}$ | $66.5 \pm 11.8$ | $55.3 \pm 2.9$ |
| 3b_vs_1hm-medium-random | $\mathbf{38.1 \pm 7.6}$ | $28.7 \pm 11.7$ | $8.1 \pm 4.7$ |

Table 2: Ablation study including alternative classifying methods (the upper part), and alternative techniques of message fusion (the lower part).

**The effectiveness of the multi-head structure** In MHCI, we use an attention-like multi-head network to predict the categories of the dataset communications. An alternative is to classify each communication to its closest category $argmin_i \, dist(m_D, \hat{m}^{(i)})$. We compare the performances using the two classification methods in the upper part of Table 2, drawing the conclusion that our multi-head structure outperforms the method of finding the closest category and acts more robustly.

**The effectiveness of linear fusion** We compare between learned linear fusion, PCA-based fusion and directly using the hidden variable shared by all communication heads. PCA-based fusion means that the fusion matrix is obtained from calculating the compression matrix given the concatentated multi-head messages (F.R.S. (1901)). And the results in the lower part of Table 2 show that learned linear fusion performs better than PCA-based fusion.

## 7 CONCLUSION

Partial observablity and Scalability are two main challenges in cooperative multi-agent RL, we look into the new area of offline communication learning that hopefully addresses the two problems to achieve higher group intelligence. However, addtional challenges come from the multi-source property of offline dataset, in which directly training communication in the end-to-end manner isn't effective enough. Therefore, we propose Multi-head Communication Imitation, aiming to combine the information in the dataset communication to boost downstream policy learning. Empirical results show the effectiveness of our algorithm under the new offline communication benchmark.

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

## A    DIFFERENT COMMUNICATION PROTOCOLS IN PREVIOUS WORKS

In Section 3, the general form of how communication works in multi-agent RL is summarized in Equation 1. We further analyze that previous works on communication can be viewed as applying different modifications on the general form.

- **Informative communication**
  $m_{ij} \in M = \mathbb{R}^d$, $d$ is the dimension of message. It is generally used in all kinds of differentiable communication learning.
- **Broadcasting communication**
  $C_i \equiv C_{i1} = C_{i2} = \cdots = C_{in} \ \forall i \in I$ (Sukhbaatar et al. (2016)).
- **Discrete communication (action communication)**
  $m_{ij} \in M = A_m = \{1, 2, \cdots, |A_m|\}$. Each agent $i$ sends the same messages to all receivers (Foerster et al. (2016)).
- **Targeted communication**
  The message follows the broadcasting communication protocol, while the policy function includes an attention module. $m_i = C(o_i), a_j = \pi_j(Attention(m_:, o_j), \tau_j)$ (Das et al. (2019)).
- **Incentive communication**
  The message directly influence the $Q$ value of each receiver. $m_{ij} \in M = \mathbb{R}^{|A|}, a_j = \arg\max \{Q(o_j, \tau_j) + m_{ij}\}$ (Yuan et al. (2022)).

## B    SUPPLEMENTARY PROOF OF THEOREM 1

In general, the proof of Theorem 1 is based on Lemma 1, which includes more efforts. In this section, we first give a proof of Theorem 1 using Lemma 1. And the residual part are all proof of Lemma 1.

**Theorem 1.** *The optimal expected return based on the Universal Communication is greater than or equal to that of dataset communications and policies.*

$$\forall (S, LC, \pi) \in G_\pi, V^*_{T_{UC}(M_\mathcal{D})}(s_{init}) \geq V_{T_{LC}(M_\mathcal{D}), \pi}(s_{init}) \tag{14}$$

*Proof.* According to Definition 1 and 2, $\forall (S, LC, \pi) \in G_\pi, C_{UC}$ dominates $C_{LC}$. Therefore by Lemma 1, we have

$$V^*_{T_{GC}(M_D)}(s_{init}) \geq V^*_{T_C(M_D)}(s_{init})$$

And apparently $V^*_{T_C(M_D)}(s_{init}) \geq V_{T_C(M_D), \pi}(s_{init})$. Therefore

$$V^*_{T_{GC}(M_D)}(s_{init}) \geq V^*_{T_C(M_D)}(s_{init}) \geq V_{T_C(M_D), \pi}(s_{init})$$

$\square$

**Lemma 1.** *If communication function $C_1$ dominates $C_2$, we have*

$$V^*_{T_{C_1}(M)}(s_{init}) \geq V^*_{T_{C_2}(M)}(s_{init}). \tag{15}$$

The following is the proof of Lemma 1. Recall that the observation function in POMDP is $O : \mathcal{S} \to \mathcal{O}$.

**Definition 3.** *The relation that $O_1$ contains more information than $O_2$ is defined as $\exists f : \mathcal{S} \to \mathcal{O}, \forall s \in \mathcal{S}(\in MDP), f(O_1(s)) = O_2(s)$.*

It can be understood as the fact that $O_1$ contains strictly more information than $O_2$.

Denote history $\tau_t = (s_1, a_1, s_2, a_2, \cdots, s_{t-1}, a_{t-1}, s_t)$, $\tau_t^{(a)} = (s_1, a_1, s_2, a_2, \cdots, s_{t-1}, a_{t-1})$ $(\forall t), \tau_t \in \mathcal{T}_t$, the probability $p(\tau)$

Denote the history under observation function $O(\cdot)$ as $O(\tau_t) = (O(s_1), a_1, O(s_2), a_2, \cdots), O(\tau_t) \in O(\mathcal{T}_t)$, the probability $p(\tau_t^O) = \sum_{O(\tau_t)=\tau_t^O} p(\tau_t)$

Denote the infostate obtained from the history $IS : O(\mathcal{T}_t) \to \Delta(\mathcal{S})$, $p_{IS(\tau_t^O)}(s) = \sum_{\tau=(\cdots, s_t), O(\tau_t)=\tau_t^O} p(\tau_t)/p(\tau_t^O)$

**Lemma 2.** $O(\mathcal{T}_t)$ *is a split of* $\mathcal{T}_t$. *If* $O_1$ *contains more information than* $O_2$, $O_2(\mathcal{T}_t)$ *is a split of* $O_1(\mathcal{T}_t)$

*Proof.*      1. $IS(O_1(\tau_t)) \subset IS(O_2(\tau_t))$.

2. Probability adds up to 1. (By taking expectation)

$\square$

Therefore define the $Split_{O_1}(O_2(\tau_t)) \in \Delta(O_1(\mathcal{T}_t))$.

**Lemma 3.** *Suppose* $\tau_t = (o_1, a_1, \cdots, o_{t-1}, a_{t-1}, o_t), \tau_t^{(a)} = (o_1, a_1, \cdots, o_{t-1}, a_{t-1})$, *if* $E_{\tau_t^{O_1} \sim Split_{O_1}(\tau_t^{O_2})} Q_{O_1}^*(\tau_t^{O_1}, a_t) \geq Q_{O_2}^*(\tau_t^{O_2}, a_t)$, *we have* $E_{\tau_t^{O_1} \sim Split_{O_1}(\tau_t^{O_2})} Q_{O_1}^*(\tau_t^{(a),O_1}, a_t) \geq Q_{O_2}^*(\tau_t^{(a),O_2}, a_t)$

*Proof.*

$$LHS = E_{o_t|\tau^{O_1}} \ E_{\tau^{O_1} \sim Split_{O_1}(\tau_t^{O_2})} Q_{O_1}^*(\tau_t^{O_1}, a_t)$$
$$\geq E_{o_t|\tau^{O_1}} Q_{O_2}^*(\tau_t^{O_2}, a_t)$$
$$= RHS$$

$\square$

**Theorem 2.** $\forall t$, $O_1$ *dominates* $O_2$, *we have* $E_{\tau_t^{O_1} \sim Split_{O_1}(\tau_t^{O_2})} Q_{O_1}^*(\tau_t^{O_1}, a_t) \geq Q_{O_2}^*(\tau_t^{O_2}, a_t)$.

*Proof.* Prove literately. Without generality, suppose all trajectories have the same length $t_{max}$, and the inequality naturally holds for the ending state.

If $E_{\tau_t^{O_1} \sim Split_{O_1}(\tau_{t+1}^{O_2})} Q_{O_1}^*(\tau_{t+1}^{O_1}, a_{t+1}) \geq Q_{O_2}^*(\tau_{t+1}^{O_2}, a_{t+1})$,

$$E_{\tau_t^{O_1} \sim Split_{O_1}(\tau_t^{O_2})} Q_{O_1}^*(\tau_t^{O_1}, a_t)$$
$$= E_{\tau_t^{O_1} \sim Split_{O_1}(\tau_t^{O_2})} \max_{a_{t+1}} Q_{O_1}^*(\tau_{t+1}^{(a),O_1}, a_{t+1}) \quad (\text{Denote } \tau_{t+1}^{(a),O_1} = Concat(\tau_t^{O_1}, a_t))$$
$$\geq \max_{a_{t+1}} E_{\tau_t^{O_1} \sim Split_{O_1}(\tau_t^{O_2})} Q_{O_1}^*(\tau_{t+1}^{(a),O_1}, a_{t+1})$$
$$\geq \max_{a_{t+1}} Q_{O_2}^*(\tau_{t+1}^{(a),O_2}, a_{t+1}) \quad (\because \text{ Lemma 4})$$
$$= Q_{O_2}^*(\tau_t^{O_2}, a_t)$$

$\square$

## C    DETAILS OF THE IMAGINARY COMMUNICATION GAME

The formulation of the imaginary communication is included in Section 4. It is named as an imaginary communication game, because it doesn't include any real-world correspondence. However, in this way we can concretely measure the influence of scalability, state space complexity and the advantage from learning communication from datasets.

The number of agents is $n$. The state space is of $p$ dimension, while the indiviual observation space is $q$. Natually, $p > q$, i.e., each agent itself doesn't observe all the information. In the implementation, we guarantee $nq > p$ and that all the concatenated observations of all agents include all the information of the state. The partial observability lies in the fact that for each agent $i$, there are $2q$ dimensions that relate to the optimal policy, but only $q$ dimensions are included in its own observation. Therefore, a communication mechanism is required to make up this lost $q$ dimensions. The

two experiments share the same network structure, but still, learning communication from scratch fails to converge to optimal when the the task become more challenging.

The network structure involves a communication module and a policy module, and both modules are 3-layer MLPs. The communication module of agent $i$ takes the individual observation $o_i$ as input and outputs the message $m_{i:}$ to each receivers. The policy module of agent $j$ takes $o_j, m_{:j}$ as input and outputs an action. All the network doesn't share parameters.

Since this environment is of horizon $T = 1$ and the reward belongs to $0, 1$, it can be modeled as a classification problem. We use BCE loss for optimization, with the data size 1000, batch size 1000 (1 step covers 1 epoch) and learning rate 0.1 in SGD optimizer. In learning communication from scratch, we optimize both the communication and policy module by the downstream BCE Loss. In the fitting dataset communication, we optimize the communication module by minimizing the MSE loss using dataset communication, while optimize the policy module by the downstream BCE loss. For the sake of fairness, we use the rule of early stopping that stops training if the evaluation accuracy doesn't increase in 30 steps.

Figure 1 is the average evaluation accuracy over 100 experiments at each value of $n, p$ with $q = 2$. In every experiment, the environment and dataset are re-initialized. In appendix, we also compare under a different $q = 3$ in Figure 6, and a different data size 10000 in Figure 7.

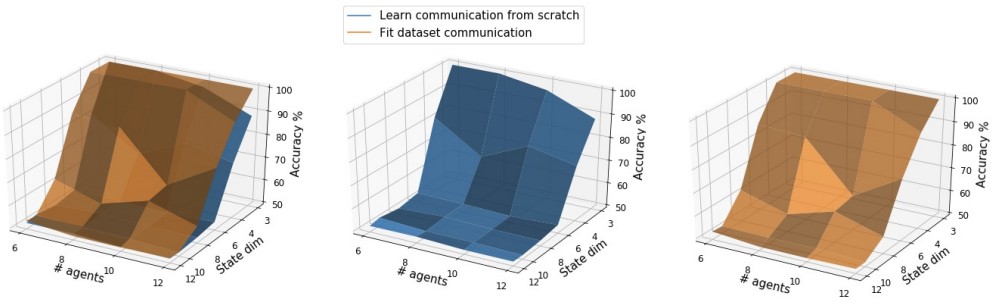

Figure 6: Evaluation accuracy with different $n, p$, given $q = 3$ (size 1000 by default).

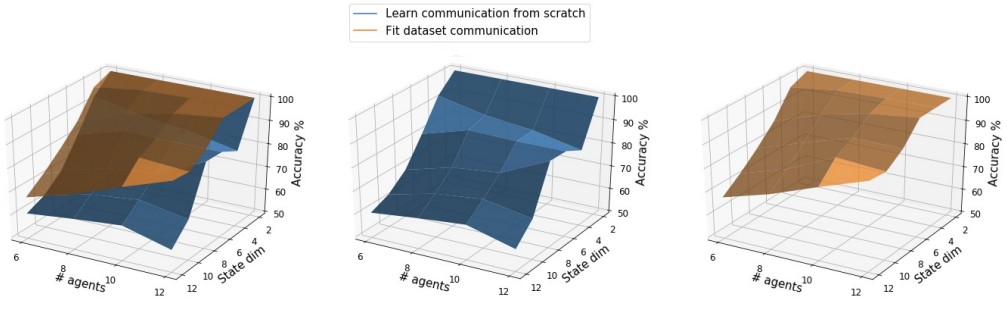

Figure 7: Evaluation accuracy with different $n, p$, given size 10000 ($q = 2$ by default).

# D IMPLEMENTATION DETAILS

## D.1 MHCI IMPLEMENTATION

Hyperparameters of MHCI and the data size in the offline communication learning benchmark are respectively presented in Table 3, 4. The implementation details of the downstream policy learning follows the same manner in ICQ's original implementation (Yang et al. (2021)).

In all the experiments, the state is the observation concatenation by default.

| Hyperparameter | Value |
|---|---|
| Communication dimensison | 3 |
| Number of attention heads | 3 |
| Batch size | 16 |
| Learning rate | 0.0005 |

Table 3: MHCI hyperparameters table.

## D.2 BENCHMARK

| Dataset | Size |
|---|---|
| Room-expert | 500 |
| Room-expert-expert-medium | 750 |
| Room-medium-random | 500 |
| All the StarCraft II tasks (medium-random) | 100 |

Table 4: The table of data size in offline communication learning benchmark.

For convenience, we use *expert* to represent manually designed datasets, which are optimal. *Medium* indicates that the dataset is collected by a learned policy and communication. And *random* is just trajectories collected by a randomly intialized policy and communication. We mix datasets from different sources and name as *expert-medium* for example, meaning the mixture of a expert dataset and a medium dataset.

Besides, for easier understanding, we mention in this paragraph the general challenge of StarCraft II tasks included in the benchmark. In previous communication-related works, the sight range of each agent is assigned to a small value, making the tasks partially observable. We also use this criterion in the added task 6o1b_vs_1r.

**Dataset generation details** In StarCraft II related tasks (3b_vs_1h1m, 1010b_vs_1r, 5z_vs_1ul, MMM, 1o2r_vs_4r and 6o1b_vs_1r), the medium dataset is generated by collecting the trajectories after 2e7 steps training of the NDQ algorithm (Wang* et al. (2020)), which means that the policy is trained to convergence. And the random dataset is collected at the first step of training, meaning that the policy is only randomly initialized. The state required by the mixing network is the concatenation of all agents' observation, rather than a predefined low-dimension state as in previous works. Because the latter requires domain knowledge which is unfeasible in the general CTDE assumption.

In the Room environment, we manually designed expert datasets, in which all the agents act perfectctly (First transmit the location to the correct agents, then move towards its own goal, as described in Section 6.1). And the communication function is designed as $m_{ij} = (1 + location) * 10$ if $i$ observes $j$'s goal, otherwise 0. And in datasets where two different expert sources are included, they're $m_{ij}^{(1)} = (1 + location) * 10, m_{ij}^{(2)} = -(1 + location) * 10$ respectively. The medium dataset is generated by offline training (training communication from scratch) on abundant expert data. And the random dataset is also collected from the trajectories of randomly initialized policies.

## E EFFECTIVENESS OF MULTI-HEAD STRUCTURE

In this section, we look into the classification performance of multi-head structure in communication learning, and show how it improves the downstream performance of the policy.

From the head classification results shown in Table 5, MHCI successfully classifies different categories of dataset communication into different heads in most datasets, which allows for accurate communication learning. And eventually the overall performance of the downstream policy is increased significantly. On contrast, in those datasets that communication classification doesn't work well (e.g. 1o10b_vs_1r, 5z_vs_1ul and Room-medium), MHCI doesn't surpass pure imitation.

(Since the second version in rebuttal updates the way of linear fusion, we need to clarify that the tendency shown in Table 5 is similar both in the old version of random linear fusion and the new version of learned linear fusion.)

| | | Multi-head | | | | Single-head | | |
|---|---|---|---|---|---|---|---|---|
| | | Head 1 | Head 2 | Head 3 | | Head 1 | Head 2 | Head 3 |
| 3b_vs_1h1m | Cate 1 | 56.2% | 40.8% | 3.0% | Cate 1 | 100% | / | / |
| | Cate 2 | 100% | 0% | 0% | Cate 2 | 100% | / | / |
| | | Score: **38.1 ± 7.6** | | | | Score: 25.7 ± 15.5 | | |
| | | Head 1 | Head 2 | Head 3 | | Head 1 | Head 2 | Head 3 |
| 1o10b_vs_1r | Cate 1 | 0% | 0% | 100% | Cate 1 | 100% | / | / |
| | Cate 2 | 0% | 0% | 100% | Cate 2 | 100% | / | / |
| | | Score: 52.2 ± 13.4 | | | | Score: **60.4 ± 6.3** | | |
| | | Head 1 | Head 2 | Head 3 | | Head 1 | Head 2 | Head 3 |
| 5z_vs_1ul | Cate 1 | 100% | 0% | 0% | Cate 1 | 100% | / | / |
| | Cate 2 | 99.8% | 0.2% | 0% | Cate 2 | 100% | / | / |
| | | Score: 47.9 ± 23.0 | | | | Score: 50.0 ± 13.5 | | |
| | | Head 1 | Head 2 | Head 3 | | Head 1 | Head 2 | Head 3 |
| MMM | Cate 1 | 0% | 41.3% | 58.7% | Cate 1 | 100% | / | / |
| | Cate 2 | 0% | 0% | 100% | Cate 2 | 100% | / | / |
| | | Score: **17.6 ± 2.9** | | | | Score: 10.0 ± 1.2 | | |
| | | Head 1 | Head 2 | Head 3 | | Head 1 | Head 2 | Head 3 |
| 1o2r_vs_4r | Cate 1 | 5.1% | 94.9% | 0% | Cate 1 | 100% | / | / |
| | Cate 2 | 0% | 100% | 0% | Cate 2 | 100% | / | / |
| | | Score: 59.7 ± 29.5 | | | | Score: **63.1 ± 15.2** | | |
| | | Head 1 | Head 2 | Head 3 | | Head 1 | Head 2 | Head 3 |
| Room-m | Cate 1 | 0% | 0% | 100% | Cate 1 | 100% | / | / |
| | | Score: 88.2 ± 4.2 | | | | Score: **95.4 ± 3.6** | | |
| | | Head 1 | Head 2 | Head 3 | | Head 1 | Head 2 | Head 3 |
| Room-ee | Cate 1 | 0% | 0% | 100% | Cate 1 | 100% | / | / |
| | Cate 2 | 0% | 100% | 0% | Cate 2 | 100% | / | / |
| | | Score: **84.5 ± 3.8** | | | | Score: 58.7 ± 0.5 | | |
| | | Head 1 | Head 2 | Head 3 | | Head 1 | Head 2 | Head 3 |
| Room-eem | Cate 1 | 0% | 0% | 100% | Cate 1 | 100% | / | / |
| | Cate 2 | 0% | 100% | 0% | Cate 2 | 100% | / | / |
| | Cate 2 | 20% | 80% | 0% | Cate 2 | 100% | / | / |
| | | Score: **66.1 ± 4.7** | | | | Score: 55.7 ± 2.5 | | |
| | | Head 1 | Head 2 | Head 3 | | Head 1 | Head 2 | Head 3 |
| Room-mr | Cate 1 | 0% | 0% | 100% | Cate 1 | 100% | / | / |
| | Cate 2 | 32.0% | 68.0% | 0% | Cate 2 | 100% | / | / |
| | | Score: **63.9 ± 6.5** | | | | Score: 61.0 ± 1.2 | | |
| | | Head 1 | Head 2 | Head 3 | | Head 1 | Head 2 | Head 3 |
| 6o1b_vs_1r | Cate 1 | 16.5% | 83.5% | 0% | Cate 1 | 100% | / | / |
| | Cate 2 | 0% | 100% | 0% | Cate 2 | 100% | / | / |
| | | Score: **47.6 ± 10.7** | | | | Score: 44.0 ± 1.4 | | |

Table 5: The predicted head of each category is shown in this table, as well as the average score of related methods. Multi-head stands for out method MHCI, and single-head means pure imitation. The dataset Room-ee is the abbreviation of Room-expert-expert. The predicted head probability is collected under one experiment after 2e5 steps, where the probability represents the proportion of communication that is classified into each head.

## F   ABLATION ON THE LEARNED LINEAR FUSION

The comparison among random fusion, learned fusion, learned fusion with learned head (used in MHCI) is shown in Table 6. Compared with the ablation study in the main text in Section 6.3, the influence of different ways of linear fusion isn't the main influencing factor of performance. Using MHCI with arbitrary linear fusion, we can obtain satisfying performance in the offline communication learning benchmark.

| | Random matrix | Learned matrix | Learned matrix + learned head (MHCI) |
|---|---|---|---|
| 3b_vs_1h1m-medium-random | $45.4 \pm 3.7$ | $38.1 \pm 3.0$ | $38.1 \pm 7.6$ |
| 1o10b_vs_1r-medium-random | $34.6 \pm 10.0$ | $48.2 \pm 4.2$ | $52.2 \pm 13.4$ |
| 5z_vs_1ul-medium-random | $62.5 \pm 22.2$ | $52.1 \pm 17.9$ | $47.9 \pm 23.0$ |
| MMM-medium-random | $29.0 \pm 4.7$ | $20.0 \pm 5.3$ | $17.6 \pm 2.9$ |
| 1o2r_vs_4r-medium-random | $65.3 \pm 13.9$ | $56.3 \pm 10.4$ | $59.7 \pm 29.5$ |
| Room-medium | $88.2 \pm 4.2$ | $88.3 \pm 1.2$ | $82.5 \pm 6.3$ |
| Room-expert-expert | $84.5 \pm 3.8$ | $81.7 \pm 3.0$ | $86.3 \pm 5.8$ |
| Room-expert-expert-medium | $66.1 \pm 4.7$ | $64.8 \pm 6.1$ | $69.5 \pm 6.7$ |
| Room-medium-random | $63.9 \pm 6.5$ | $62.8 \pm 5.0$ | $67.2 \pm 6.4$ |
| 6o1b_vs_1r-medium-random | $47.6 \pm 10.7$ | $39.0 \pm 4.9$ | $45.0 \pm 5.7$ |

Table 6: The performance comparison of different ways of linear fusion, normalized by the score of the expert (from 0 to 100). The results are averaged over three random seeds with standard deviation.

## G    THE EFFECT OF COMBINING DIFFERENT COMMUNICATION PROTOCOLS

We look into how MHCI combines different communication protocols and improve the overall performance. We specifically design the dataset including multiple sources that are valuable in different aspects, and therefore combining all of them, rather than learning one of them, is essential for good performance.

In detail, we modify the Room-medium dataset by masking half of the senders' messages, and leaving the trajectories the same. In 50% data, we mask the first 4 agents' sent messages, and in the remaining 50% data, we mask the last 4 agents' sent messages (8 agents in all). Therefore, the communication learning algorithm needs to combine the experiences of how the first 4 agents should communicate, and also how the last 4 agents should communicate. The experiment results shows that, under such dataset that requires combining experiences, MHCI gains advantage by combining different communication protocols, and exceeds learning from scratch.

| 50% mask 1-4 50% mask 5-8 | 100% mask 1-4 | Not masked | Learn communication from scratch |
|---|---|---|---|
| $82.0 \pm 11.4$ | $69.5 \pm 2.3$ | $88.3 \pm 1.2$ | $74.2 \pm 3.5$ |

Table 7: The experiment results when the optimal communication in the dataset is masked partially, showing that MHCI combines experience from different sources of communication protocols.

## H    COMPARISON WITH PREVIOUS ONLINE COMMUNICATION LEARNING ALGORITHMS

We compare MHCI with VBC (Zhang et al. (2019)) under 3b_vs_1h1m-medium-random in Figure 8. VBC fails to converge because it isn't designed for the offline setting.

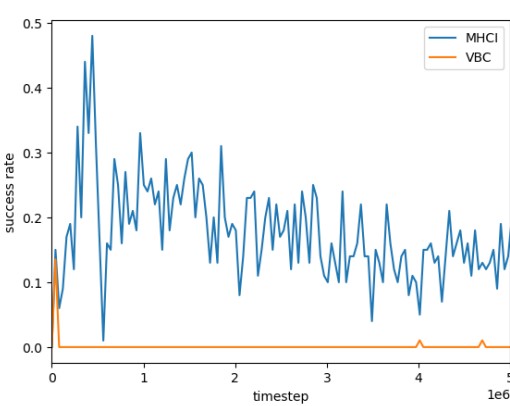

Figure 8: Learning curve on 3b_vs_1h1m-medium-random.

