# OpenReview forum: "Offline Communication Learning with Multi-source Datasets"
_ICLR.cc/2023/Conference — Submitted to ICLR 2023_

### Official Review · Reviewer_UfVH · 2022-10-24

**Confidence:** 5
**Clarity, Quality, Novelty And Reproducibility:** See Strength And Weaknesses.
**Correctness:** 3
**Technical Novelty And Significance:** 2
**Empirical Novelty And Significance:** 2
**Recommendation:** 5

**Strength And Weaknesses:**

Pros:
1. The motivation is clear.
2. The paper is well-written and organized.
Cons:
1. More results on more challenging datasets are needed to verify the superiority of the proposed method.
2. Some related works are missing, e.g., Multi-caption Text-to-Face Synthesis: Dataset and Algorithm.

**Summary Of The Paper:**

The authors analyze the new challenges in offline communication learning, and introduce a benchmark of offline communication learning which contains diverse tasks. The author propose an effective algorithm, Multi-head Communication Imitation (MHCI), which aims to address the problem of learning from single-source or multi-source datasets, and our method shows superior outperformance in various environments of our benchmark.

**Summary Of The Review:**

See Strength And Weaknesses.

---

> ### Author Response · Authors · 2022-11-13
> **Reply**
>
> Thanks for the review. The following are the reply to the weaknesses.
>
> 1. Your advice is meaningful. We also emphasize the importance of a diverse benchmark of challenging tasks, and it also requires much effort, which would be our future work.
>
> 2. We add the citation in Section 2.

---

### Official Review · Reviewer_zXRw · 2022-10-24

**Confidence:** 4
**Correctness:** 3
**Technical Novelty And Significance:** 3
**Empirical Novelty And Significance:** 3
**Recommendation:** 5

**Clarity, Quality, Novelty And Reproducibility:**

*Clarify: This paper is well-written and overall clear.
*Novelty: The problem of this work is somewhat novel, but the design of the network seems to be similar to prior work.
*Reproducibility: code is provided

**Strength And Weaknesses:**

**Strength**
- This paper is relatively easy to follow.
- The proposed network seems to have better performance compared to the chosen baselines.

**Weakness and Questions**
- How are the different datasets generated (e.g. export, medium, etc)? Does the model still work well if the communication protocol changes completely? More details should be provided.
- The method should also be evaluated under more combinations of communication messages to understand the performance related to the difference in the multi-source data. How do combining different trajectories in different datasets and different communication protocol affects the performance of the proposed algorithm? Can the algorithm learn to use communication strategy learned from some trajectories to a different trajectory to improve performance? More ablation would be helpful here.
- In the evaluation section, it would be a good idea to provide the performance of the different benchmarks under the online learning case. For example, comparing the proposed method to online techniques (e.g. I2C) in MMM.
- In the motivating example, more details should be provided in terms of how the two models (learning communication from scratch and feed dataset communication) are trained.


**Summary Of The Paper:**

This work investigates learning the communication strategy in MARL from offline communication datasets. The authors proposed a multi-head attention method to encode the messages to support learning data from multiple sources. The proposed method is evaluated using a custom window environment and a custom StarCraft environment.

**Summary Of The Review:**

This work is a looking at the communication problem in offline MARL. It tries to learn from communication data that is multi-sourced. The proposed method seems to be effective and the problem is relatively new; however, the challenge of the research problem seems limited which limits the novelty of the proposed method.

---

> ### Author Response · Authors · 2022-11-13
> **Reply**
>
> Thanks for the review. Here are the reply to the weakness and questions.
>
> 1. We add dataset generation details in Appendix D.2.
>
> 2. We think that the main results in Table 1 already show the effectiveness of our method under 1-3 sources and levels among random, medium and expert, and more than 3 sources may be left to future works. But we agree that it’s important to analyze the effect of combining different experience, which is absent in the paper. Therefore, we carry out the experiment on a designed dataset consisting of two sources of communication protocol, but requires combining those two sources in order to learn an optimal policy in Appendix G. We’re glad if you could check this part and tell us whether it clear up your doubt.
>
> 3. Thanks for the advice about providing the comparison with other online communication algorithms. We provide a brief comparison with VBC in Appendix H. There’re some difficulties of providing more comparison: 1. Many works don’t support Starcraft II environment (e.g. I2C, Tarmac, etc.); 2. Without a backend offline RL algorithm, the score is always very low, which means that we need to manually modify their algorithms.
>
> On the other perspective, existing communication researches are designed for online settings (how to effectively find the optimal communication), but the offline setting faces completely different challenges (how to combine existing communications).
>
> 4. In learning communication from scratch, we optimize both the communication and policy module by the downstream BCE Loss. In fitting dataset communication, we optimize the communication module by minimizing the MSE loss using dataset communication, while optimize the policy module by the BCE loss. (We also add those descriptions into the appendix)

---

### Official Review · Reviewer_gcz9 · 2022-10-24

**Confidence:** 4
**Correctness:** 3
**Technical Novelty And Significance:** 3
**Empirical Novelty And Significance:** 3
**Recommendation:** 3

**Clarity, Quality, Novelty And Reproducibility:**

- Clarity: The paper is clear to me.
- Quality: There is still much space for improvement in terms of the motivation/justification of the core idea, the presentation of graphs, and the notations.
- Novelty: The idea is interesting. However, I think some justifications are needed for this setting.
- Reproducibility: the authors provide part of the source code.


**Strength And Weaknesses:**

Pros:

- This paper takes on an interesting problem in MARL: the combination of communication and offline MARL.
- The experiments are completed.

However, I have several concerns here:
- To my understanding, this paper is mainly to learn a universal communication protocol to achieve higher accuracy. This point is interesting from the perspective of MARL communication but is not adequate. Since we can just view this problem as simple supervised learning. End-to-end communication offline learning is more attractive.
- Why do we need an offline MARL dataset with communication messages? The objective of offline RL/MARL is to obtain good policies from the dataset for direct deployment. We can have an offline MARL dataset by running communication but discarding communication messages. Since the messages are mainly from the observations of each agent. Will agent communication during deployment? If not, we can mimic such a dataset for deployment. If so, we can also have a communication mechanism inside the policy learning without mimicking the multi-source messages.
- In Fig.1, Left = Mid + Right. No need to plot redundant figures.


**Summary Of The Paper:**

This paper mainly considers communication in offline multi-agent reinforcement learning (MARL). The authors propose a new benchmark that contains a set of offline MARL communication tasks with single/multi-source datasets. The “multi-source” means that the messages are from different communication protocols. The authors show that there exists a universal communication protocol that can represent all sources. Also, the authors propose a multi-head structure for communication Imitation learning. Experiments show a good performance in a few scenarios in SMAC and Room.

**Summary Of The Review:**

I like the idea of incorporating communication into offline MARL. However, I think the authors should improve the current version.

---

> ### Author Response · Authors · 2022-11-13
> **Reply**
>
> Thanks for the review. The following are the reply to your concerns.
>
> 1. It’s completely true that offline communication learning can be viewed as a pure supervised learning process. We put emphasis on this topic rather than end-to-end learning because end-to-end communication learning is fully studied in online communication-related works (e.g. Tarmac, NDQ, etc.), and the offline setting doesn’t pose extra challenges to end-to-end communication learning. Interestingly, widely-used tasks in communication-related works are easy in the offline setting because cooperation is already in the dataset (Further discussion can be found in the introduction). As a result, under the offline setting, we can actually solve harder tasks, which are unsolvable online. That’s the main motivation of our work.
>
> 2. We admit that in most cases, dataset with only trajectories are enough for optimal policy learning, as communication can be learned directly from the policies. But in our work, a more extreme situation is considered, that the state space is much too large and normal RL algorithms would fail to obtain the optimal policy. It is common in real world (e.g. image inputs, large systems have large state space). Compared to the large state space, the communication is in fact a condensed information. Therefore, imitating the optimal communication would help in such cases.
>
> 3. Thanks for your advice. We have changed Figure 1.

---

### Official Review · Reviewer_311E · 2022-10-25

**Confidence:** 4
**Correctness:** 3
**Technical Novelty And Significance:** 3
**Empirical Novelty And Significance:** 2
**Recommendation:** 3

**Clarity, Quality, Novelty And Reproducibility:**

The paper is well-written in composition and order with proper mathematical theories.

It is meaningful and novel in that the paper proposes a new kind of multi-agent offline RL environment, with agents with communication.
The motivation is, if the method learns the communication of the dataset, it will increase the performance rather than learning from scratch without the dataset. But it seems natural to perform better with using more information. Of course, the comparing result with simple imitation learning is included, and more powerful motivation (e.g. the existing imitation learning structures are not suitable for learning communication) is needed.

Typo: page2 DAIL, RAIL -> DIAL, RIAL / figure2 $h_j^t$


**Strength And Weaknesses:**

Strengths

This paper presents a new problem of how to learn the offline multi-agent RL datasets with communication and proposes an algorithm based on a multi-head attention structure to solve it.
The paper shows the performance of the proposed method by creating a new benchmark, and it shows higher performance than the existing offline RL algorithm.
Mathematical concepts explain well how the proposed algorithm can get a high return in an environment where datasets are generated from various distributions.

Weaknesses

It seems more necessary to analyze whether the proposed multi-head structure is suitable for problem-solving. When learning based on various datasets, it is necessary to analyze whether each head is learned diversely as intended and to provide evidence of performance increase.

There is also a lack of explanation on how to obtain the dataset. There will be a variety of ways to design the communication function. Since the created message is included in the dataset, the algorithm performance may change significantly depending on the algorithm that created the dataset.

Since it is an algorithm designed to learn datasets made from various distributions well, it is also effective to show the performance change according to the number of distributions included in the dataset.

The explanation and backgrounds for linear fusion are poor. The multi-head attention structure is designed to learn messages from various distributions separately, but it is too naive to randomly mix all heads in actual use. In the process of randomly mixing multi-heads, the basis of the structure seems to disappear. The possibility of a structure that uses learned multi-head more efficiently is open.
The explanation of the base algorithm ICQ seems to be poor, and it is necessary to explain why this algorithm is chosen.


**Summary Of The Paper:**

The paper is about how to learn offline environments that can communicate between agents in multi-agent environments. Each agent learns a function that generates messages from their observations, based on the messages in the datasets. The communication function is learned with a multi-head attention structure, and in the evaluation, agents transfer messages derived from a linearly random mixing of messages learned from each head. Experiments have been performed in environments that require intensive communication, based on Room and StarCraft 2. The proposed structure shows higher performance than learning communication without message information in datasets or imitation learning without a multi-head attention structure.

**Summary Of The Review:**

This paper is meaningful in that it attempts to approach a new kind of multi-agent offline RL environment. However, the paper is poor at showing the process of generating the datasets, showing the performance of the proposed method(more evaluation and ablation that shows the advantages of the structure should be provided), and the lack of explanation and analysis of linear fusion.

---

> ### Author Response · Authors · 2022-11-13
> **Reply**
>
> Thanks for the review. The following replies for the weaknesses in order.
> 1. We agree that the effect of the multi-head structure is not mentioned in the paper. We conduct additional analysis on what the respective heads learn, and the effect of such process in Appendix E. It shows that our method identifies different categories of communication and gains advantages in the downstream policy training.
>
> For example, the proportion of predicted head of Room-expert-expert is:
>
> | | Head1 | Head2 | Head 3 |
> | ---| ------- | ------- | ------ |
> | Cate 2 |     0%    |   0%      | 100% |
> | Cate 1 | 0% | 100% | 0% |
>
> | | Head1 | Head2 | Head 3 |
> | ---| ------- | ------- | ------ |
> | Cate 2 |     58.2%    |   40.8%      | 3% |
> | Cate 1 | 100% | 0% | 0% |
>
> The result of all datasets are in Appendix E.
>
> 2. We analyze existing researches in communication in Appendix A and concludes that they can all be unified as the general form in Equation (1), while taking respective inductive bias (e.g. locality, attention structure, etc.). Our method MHCI is based on the general form of communication, and therefore theoretically applicable to all kinds of communication protocols. In the experiment part, we do neglected datasets generated by other communication protocols, which needs to be improved in future works. The difficulty that prevents us from providing those experiments is that: 1. Our algorithm is based on policy gradient, some existing works are value-based ones; 2. Some classical environments in previous works are too simple in the offline setting (e.g. traffic system, predator prey).
>
> 3. In the experiment section, we show the results of our method under different datasets, consisting of 1, 2 and 3 distributions. The distributions also range from medium to expert level. We believe that those combinations are an indicator of effectiveness.
>
> 4. In the main results in Table 1, we test on Room-medium, Room-expert-expert, Room-expert-expert-medium, ranging from 1, 2, 3 sources.
>
> 5. We appreciate your advices on the linear fusion part. Therefore we make an update on the linear fusion part in the newest version. In this version, we replace random linear fusion with learned linear fusion that is automatically decided in the training. The details are updated in Section 5.2.
>
> The reason we chose ICQ is that, when we start this project, there’re only a few works on offline MARL (mentioned in Section 2). ICQ performs well in StarCraftII-related tasks.

---

### Decision · Program_Chairs · 2023-01-20

**Decision:**

Reject

**Justification For Why Not Higher Score:**

All the reviewers agreed to reject this paper. They pointed out several shortcomings of the paper, the authors tried to address those during the rebuttal but the initial paper is not ready for the publication. The paper seems to be rushed. This paper would benefit another round of reviews with all the changes the authors propose.

**Justification For Why Not Lower Score:**

There is no lower score.

**Metareview: Summary, Strengths And Weaknesses:**

## Summary

This paper considers how to incorporate communication between offline MARL agents. The paper proposes a communication mechanism that makes use of multihead atttention and provide experimental results on the Room and SC2 environments.

I will list some of the strengths and weaknesses of the paper as pointed out by the reviewers:

## Strengths
- This paper is well-written and easy to follow.
- Studies and interesting problem.

## Weaknesses
-  Lack of explanation on how to obtain the dataset and how they are generated.
- The poor explanation and backgrounds for linear fusion.
- The lack of analysis on whether the proposed multi-head structure is suitable for problem-solving.
- More results on more challenging datasets are needed to verify the superiority of the proposed method.
- Some related works are missing, e.g., Multi-caption Text-to-Face Synthesis: Dataset and Algorithm.

*Decision:*
The authors did a good job at addressing some of the concerns raised by the reviewers. However, it seemed like the initial version of the paper was not completely ready for the publication and a bit rushed. The justification of the proposed approach was not completely clear. I recommend the authors to take the feedback provided by the reviewers into consideration and submit to a different venue.